# Genetic characterization of a novel pheasant-origin orthoreovirus using Next-Generation Sequencing

Yi Tang[1], Haiyang Yu[2], Xiaoning Jiang[1], Endong Bao[2], Dong Wang[2], Huaguang Lu[3]*

1 College of Animal Science and Technology, Shandong Agricultural University, Taian, Shandong Province, China, 2 Tianjin Ringpu Bio-Technology Co, Ltd., Tianjin, China, 3 Wiley Lab / Avian Virology, Animal Diagnostic Laboratory, Department of Veterinary and Biomedical Sciences, The Pennsylvania State University, University Park, PA, United States of America

* hxl15@psu.edu

**Data Availability Statement:** All relevant data are within the paper and its Supporting Information files.

## Abstract

A field isolate (Reo/SDWF /Pheasant/17608/20) of avian orthoreovirus (ARV), isolated from a flock of game-pheasants in Weifang, Shandong Province, was genetically characterized being a field variant or novel strain in our recent research studies in conducting whole genome sequencing by using Next-Generation Sequencing (NGS) technique on Illumina MiSeq platform. Among a total of 870,197 35-151-mer sequencing reads, 297,711 reads (34.21%) were identified as ARV sequences. The *de novo* assembly of the ARV reads resulted in generation of 10 ARV-related contigs with the average sequencing coverage from 1390× to 1977× according to 10 ARV genome segments. The complete genomes of this pheasant-origin ARV (Reo/SDWF /Pheasant/17608/20) were 23,495 bp in length and consist of 10 dsRNA segments ranged from 1192 bp (S4) to 3958 bp (L1) encoding 12 viral proteins. Sequence comparison between the SDWF17608 and classic ARV reference strains revealed that 58.1–100% nucleotide (nt) identities and 51.4–100% amino acid (aa) identities were in genome segment coding genes. The 10 RNA segments had conversed termini at 5' (5'–GCUUUU) and 3' (UCAUC–3') side, which were identical to the most published ARV strains. Phylogenetic analysis revealed that this pheasant ARV field variant was closely related with chicken ARV strains in 7 genome segment genes, but it possessed significant sequence divergence in M1, M3 and S2 segments. These findings suggested that this pheasant-origin field variant was a divergent ARV strain and was likely originated from reassortments between different chicken ARV strains.

## Introduction

Viruses of the family *Reoviridae* have segmented dsRNA genomes and are classified into two subfamilies comprising a total of 15 genera [1]. The genus *Orthoreovirus* belongs to the subfamily *Spinareovirinae*, which currently has five species, including Mammalian orthoreovirus, Baboon orthoreovirus (BRV), Reptilian orthoreovirus (RRV), Nelson Bay orthoreovirus

**Funding:** The author(s) received no specific funding for this work.

**Competing interests:** The authors have declared that no competing interests exist.

(NBV), and Avian orthoreovirus (ARV) [2, 3]. ARV is non-enveloped virus with icosahedral symmetry and contains a surface protein arranged in a double shell [4]. Virus particles have an average size of 70–80nm and are wrapped around 10 genomic fragments. Depending on their movement in electrophoresis, linear genomic fragments were molecularly divided into three different groups, including three large fragments (L1-L3), three medium fragments (M1-M3) and four small fragments (S1-S4) [5, 6]. Besides the tricistronic S1 segment, all ARV genome segments are monocistronic and all genome segments encode 8 structural proteins (λA, λB, λC, μA, μB, σA, σB and σC) and 4 nonstructural proteins (μNS, p10, p17 and σNS) [7]. Each ARV coding gene was flanked by 5' and 3' non-translated regions and the first seven bases (5'-GCUUUUU) and the last five bases (UCAUC-3') of segment termini were found are highly conserved in known ARV strains [8].

ARV is a highly contagious virus involved in a variety of disease conditions or syndromes in poultry, of which viral arthritis/tenosynovitis is the most classic leg lameness or weakness seen in ARV-affected young broiler chickens [3, 9]. Since the 1980s, with the rapid development of the modern poultry industry, new symptoms or newly observed disease problems associated with ARV infections have been continuously reported, such as runting-stunting syndrome (RSS) [10], immunosuppression [11], hepatitis [12], malabsorption/maldigestion syndrome [13], respiratory disease [14], and enteric disease [15]. In recent years in Pennsylvania, commercial poultry flocks suffered viral arthritis/tenosynovitis have been increasingly diagnosed and field variant strains of the newly emerging ARVs were confirmed as the causative agent [16, 17]. During the same period, highly pathogenic ARV variants emerged in turkey in Midwestern United States [18, 19], as well as in other countries [18–20]. Similarly, an increasing number of cases of arthritis in broilers caused by ARV has occurred in China, and the ARV field variants remain higher genetic diversity and virulence in flocks, which caused considerably economic losses in the poultry industry. Most of the emerged variants showed common features of genome segments reassortments with historical ARV strains and high genetic diversity in σC genes [21].

Many research studies on ARV infections in various avian species, especially domestic poultry, have been well documented, such as broiler breeders [22], layer breeders [23], broilers [24], geese [25, 26], turkeys [27, 28], ducks [29–31], pigeons [32], and quails [33–35]. Although ARV transmissions commonly occur within and between avian species, the importance of wild birds as reservoirs of ARV transmission source to domestic poultry infections was not well studied until recent years due to the difficulty of wild bird sampling [36, 37]. Until now, there were only two pheasant cases of ARV infections were reported in 1990s, one case ARV strain was associated with hepatopathy symptoms and the other was associated with tenosynovitis [38], however, there is no previous report of pheasant being infected by ARV in China. In the present study, we report our findings of isolation and full-genome characterization of a novel pheasant-origin ARV field variant strain.

## Materials and methods

### Virus isolate

The pheasant ARV isolate (Reo/SDWF /Pheasant/17608/20) in this study was isolated from tendon tissue of a pheasant case with hepatopathy symptoms at 2–4 weeks of age. The ARV isolation was made in LMH cell cultures and produced giant or bloom-like cytopathic effect (CPE) cells, which were characteristic to ARV and confirmed positive for ARV by fluorescent antibody (FA) test using ARV conjugated antibody (ID No. 680 VDL 9501, NVSL, Ames, IA, USA) which described in our previous study [39]. The ARV cell culture material was subsequently tested negative for other avian viruses which could cause the pheasant leg lesions such

as avian influenza virus, Newcastle disease virus, fowl adenovirus type 1 and rotavirus. This pheasant ARV was propagated in LMH cell cultures, tittered as $10^{8.5}$ TCID$_{50}$/mL, aliquoted and stored at -80˚C freezer for this study.

## RT-PCR and Sanger sequencing

Viral RNA extraction of the pheasant ARV was performed by using MiniBEST Universal RNA Extraction Kit (TaKaRa, Dalian, China) per the manufacturer's instructions. Conventional RT-PCR reaction was carried out with P1/P4 primers which corresponding to 3' end of S1 segment (σC gene) of ARV [39] by using the One Step RT-PCR Kit Ver.2 (TaKaRa, Dalian, China). RT-PCR products were observed by 1% agarose gel electrophoresis and the 1088bp band was purified by using gel extraction kit (OMEGA, D2500-02 Gel Extraction Kit, USA) following the manufacturer's instructions. The concentration of the purified DNA was confirmed by using a NanoDrop™1000 (DeNovix DS-11, USA) spectrophotometer and then submitted to Personalbio for Sanger sequencing.

## Next-generation sequencing

NGS was carried out on a Miseq platform. Total RNA samples were processed by MiniBEST Universal RNA Extraction Kit (TaKaRa, Dalian, China) to build cDNA library. Briefly, the total RNA was fragmented into small pieces using magnesium divalent cations under elevated temperature [40]. The cleaved RNA fragments were copied into the first strand cDNA using reverse transcriptase (Invitrogen, Grand Island, NY, USA) and random primers. The second strand cDNA synthesis was followed using DNA polymerase I and RNase H but without the initial poly A enrichment step. Then these cDNA fragments were assessed by Direct Detect™ bioanalyzer system (DeNovix DS-11, USA) to test the fragments distribution. Thereafter, the prepared cDNA library was loading on the Miseq sequencer to get the raw NGS reads.

## *De novo* assembly of viral genome

The CLC Genomics Workbench V7.5.2 software (QIAGEN, Boston, MA, USA) was used for NGS raw data De novo assembling. Firstly, sequencing adaptors were trimmed off and contaminants sequences mapped to rRNA and mRNA reads were removed. The rest of clean reads were processed by software to get contigs. All ARV-related contigs were selected to build the full-length genome of ARV based on the BLASTN searching result. Each assembled segment was further upgraded by mapping back all NGS raw reads to the ARV-related contigs, and the consensus sequences were considered as the complete ARV genome.

## Sequence analyses

The modules of EditSeq and MegAlign of DNASTAR Lasergene 12 Core Suite (DNASTAR, Inc. Madison, WI, USA) were used for viral open reading frames (ORFs) prediction, amino acid (aa) translation, sequence alignment, and pair-wised sequence comparison. An online search program (http://blast.ncbi.nlm.nih.gov/Blast.cgi) identified the highest similarities between the studied ARV genome segment and the published sequences.

Sequencing coverage, mapped reads, and intra-host single-nucleotide variants (iSNVs) of each assembled contigs were calculated and visualized by CLC Genomic Workbench V7.5 software (QIAGEN, Boston, MA, USA). Phylogenetic analysis of genome segments were carried out by using the neighbor-joining method in MEGA CC program [41] and the bootstrap validation method with 1000 replications. The visualization of genome alignment was performed using the mVISTA (http://genome.lbl.gov/vista/mvista/submit.shtml) and the scale

sequence was using studied pheasant-origin ARV genome. Genome sequences of 13 ARV reference strains were retrieved from Genbank for sequence comparisons (S1 Table).

## Results

### Sanger sequencing

One-step RT-PCR was used to test the pheasant ARV viral RNA using specific primers (P1/P4) based on the S1 gene. As a result, the 1088bp PCR product was successfully amplified, and sanger sequencing results of showed the pheasant ARV strain (MZ561700) has about 88% nucleotides homology to those of other novel GoAstV strains, ARV strain in GenBank (L07069). In the next-generation sequencing analysis, the contigs of ARV were clearly mapped, whereas draft contigs to other-related viruses were not mapped.

### Analyzing the NGS raw data

A total of 842,235 sequencing reads of 35-151-mer were generated on Miseq sequencer from extracted viral total RNA. The final NGS sequence data from the viral stocks output file in fastq format was 79.3Mb in size. Low-quality reads, trim poly-T tails and adapter sequences were processed by quality control (QC) filters of the Miseq platform for removal, A further screening of the NGS reads was carried out to remove the non-research target readings that were similar to mRNA or rRNA sequences. As a result, 420,914 reads (48.37%) were identified to be the chicken rRNA source and 109,731 reads (12.61%) to be the chicken mRNA source (Fig 1A). The residual 339,525 clean reads were further analyzed using a BLASTN procedure, which further divided into no hits group (41,814 reads, 4.81%) and orthoreovirus group (297,711 reads, 34.21%) (Fig 1A).

### De novo assembly of viral genomes

After the assembly of no hits reads and orthoreovirus by *de novo* assembler of CLC Genomics Workbench software, a total of 72 contigs were generated with length from 150nt to 3958nt. The total reads counts of assembled contigs were found from 2 to 102,404 and the average coverage were from 1.08× to 6,506.34×. By searching online using BLASTN, 10 of the 72 contigs (Table 1) were identified as ARV-related contigs, ranging in length from 1192nt to 3958nt. The highest similarity search of 10 ARV contigs in Genbank revealed that all contigs had different homology (88.0%-96.6%) to other published reference ARV strains (Table 1). The initial alignment of assembled contigs and most homology reference ARV segments indicated the full-length of 10 ARV segments including the 5' and 3' termini were individually targeted by 10 ARV contigs.

### Sequencing coverage

By mapping back all NGS raw reads to the 10 assembled ARV contigs, the mapped reads and sequencing coverage of different genome segments were finally obtained. Although the mapped reads of each segment were various from 14,190 to 60,810, they were still positively correlated with the contig length (Table 1, Fig 1B). Sequencing coverage for each genome fragment averaged from 1390× to 1977×, indicating that there is sufficient redundancy to identify the PA136491 pheasant ARV genome. The sequencing coverage of each ARV fragment is illustrated by the wave-chart (Table 1, Fig 1C), with a peak (2937×) appearing in the L3 fragment.

Sequencing coverage and intra-host single nucleotide variants (iSNVs) were further analyzed using the resequencing analysis module of the CLC Genomic Workbench software. As a result, regions of high sequencing coverage were found throughout the ARV genome (Fig 1C)

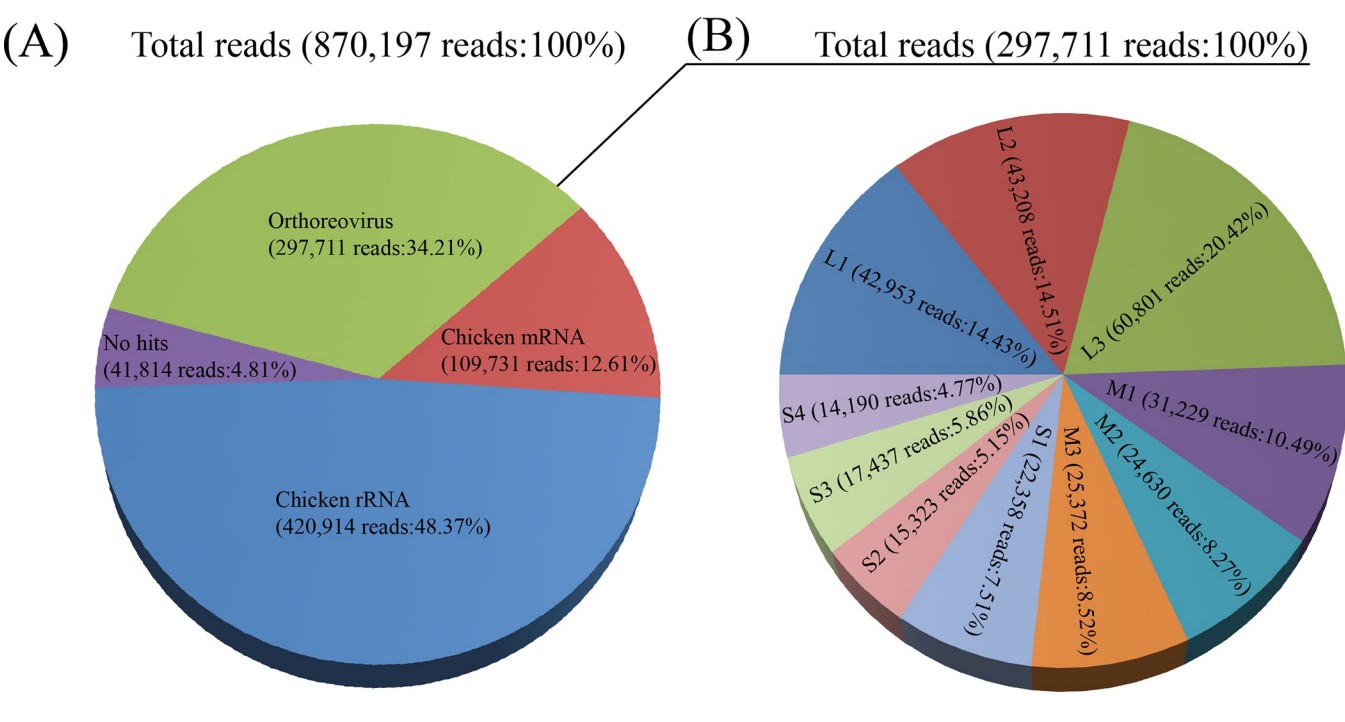

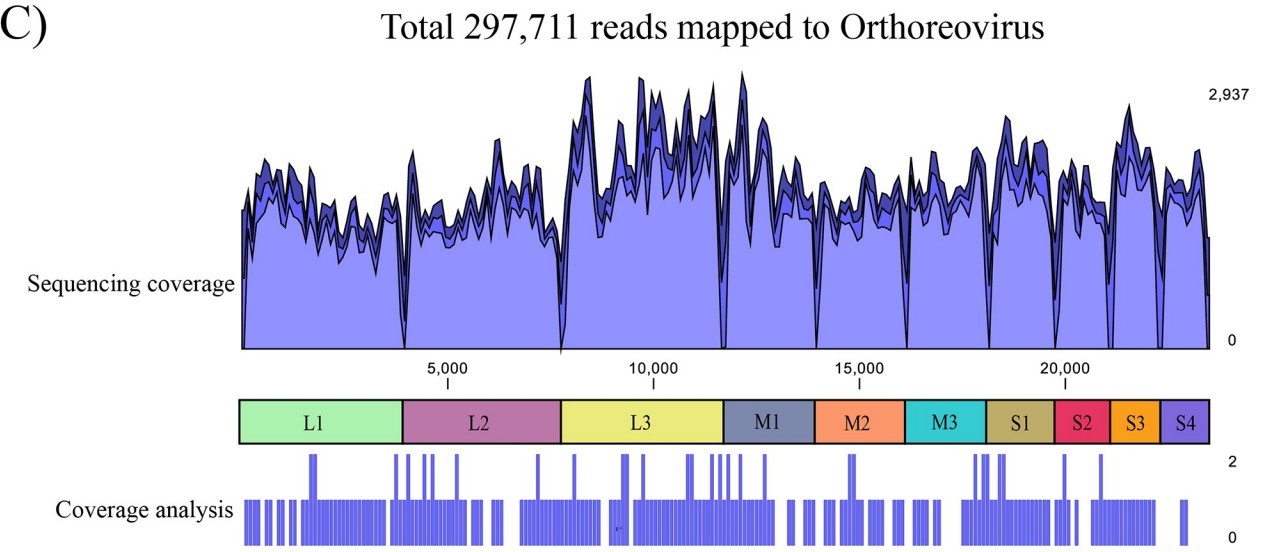

**Fig 1. The illustrations of the homology search results for NGS reads and the sequencing coverage analysis.** A: Total NGS reads homology search result; B: The remapping of NGS raw reads to ARV 10 segments; C: The result of sequencing coverage analysis.

by setting above 10 bp as the date aggregation value and a total of 7 iSNVs were determined in L1 and M2 segments (Table 1) by setting the sequencing error correction value of 0.4%.

## General genome information

Finally, the complete sequences of 10 fragments of the pheasant antiretroviral genome were obtained and deposited into Genbank (MZ561694 to MZ561703). The SDWF17608 ARV genome fragment ranges from 1192bp (S4) to 3958bp (L1), with a total size of 23,495bp. The

**Table 1. General genome features of a pheasant-origin avian orthoreovirus (ARV), the novel field variant of pheasant ARV (Reo/SDWF/Pheasant/17608/20).**

| Contig Length (bp) | Contig name | Highest similarity ARV strain in GenBank | Identities (%) | SNVs | Mapped reads | Average coverage | Encoded protein | Length of the(bp) | | |
|---|---|---|---|---|---|---|---|---|---|---|
| | | | | | | | | 5' end | ORF | 3' end |
| 3958 | L1 | 919 strain; segment L1, lambda A gene (AY641739) | 96.6 | 1 | 42953 | 1390 | λA(core shell) | 20 | 3882 | 56 |
| 3829 | L2 | AVS-B strain; segment L2, lambda B gene (FR694192) | 96.3 | 0 | 43208 | 1439 | λB(core RdRp) | 13 | 3780 | 36 |
| 3907 | L3 | 138 strain; segment L3, lambda C gene (EU707937) | 93.6 | 0 | 60801 | 1977 | λC(core turret) | 12 | 3858 | 37 |
| 2283 | M1 | 138 strain segment, M1, muA gene (AY557188) | 88.2 | 0 | 31229 | 1730 | μA(core NTPase) | 13 | 2199 | 72 |
| 2158 | M2 | Reo/PA/Broiler/05682/12 strain, segment M2, muB gene (KM877329) | 99.4 | 6 | 24630 | 1445 | μB(outer shell) | 29 | 2031 | 98 |
| 1996 | M3 | 1017−1 strain, segment M3, muNS gene (AY573905) | 99.5 | 0 | 25372 | 1564 | μNS(NS factory) | 24 | 1908 | 64 |
| 1644 | S1 | Somerville 4 strain, segment S1, p10, p17 and sigma C genes (L07069) | 88.0 | 0 | 22358 | 1721 | p10(NS FAST) | 22 | 300 | 33 |
| | | | | | | | p17(NS other) | | 441 | |
| | | | | | | | σC(outer fiber) | | 981 | |
| 1324 | S2 | 526 strain, segment S2, sigma A gene (KF741703) | 89.1 | 0 | 15323 | 1455 | σA(core clamp) | 15 | 1251 | 58 |
| 1202 | S3 | Reo/PA/Broiler/05682/12 strain, segment S3, sigma B gene (KM877333) | 91.5 | 0 | 17437 | 1710 | σB(outer clamp) | 30 | 1104 | 68 |
| 1192 | S4 | 526 strain, segment S4, sigma NS gene (KF741705) | 94.3 | 0 | 14190 | 1476 | σNS(NS RNAb) | 23 | 1104 | 65 |

GC content displays variation between different genome segments, from 47% to 52%. There were 12 viral proteins encoded by one tricistronic segment (S1) and nine monocistronic segments. ORFs range in length from 3882 bp (λA) to 300 bp (p10), which were similar to the general characteristics of published ARV strains. Although the size of most proteins was identical between this pheasant ARV and the reference ARV strains, but the pheasant ARV nonstructural protein p10 encoding gene on S1 segment showed heterogeneity.

The p10 gene of the pheasant ARV was 300bp (100 aa) in length, which was larger than that of chicken ARV strains of PA05682 (291 bp, 97 aa), S1133 (297 bp, 99 aa), 1733 (297 bp, 99 aa) and 138 (297 bp, 99 aa) (Fig 2); whereas the p10 gene was the same between the pheasant ARV and the PA22342 turkey ARV. The untranslated regions (UTRs) were sited at the 5' and 3' termini of each genome segment with length of 12-30bp (5' UTRs) and 33-98bp (3' UTRs). The highly conserved terminal sequences of 5' UTR (5'-GCUUUU-3') and 3' UTR (5'-UCAUC-3') of the pheasant ARV were undistinguishable from other ARV reference strains, but distinguishable from the non-ARV reference strains (Table 2).

## Sequence comparisons

In homologs comparison of the nucleotide (nt) and aa sequences, we found that the pheasant ARV had various similarities ranging from 42.5% nt to 99.6% nt and 29.7% aa to 99.7% aa (Table 3), which were obtained in pairwise comparisons with six ARV reference strains and seven non-ARV orthoreovirus strains (S1 Table). In addition, the comparison results of σC encoding genes revealed that was the most divergent gene between the SDWF17608 and other reference ARV strains by showing the extremely low nt and aa identities (nt: 42.5–64.1%; aa:

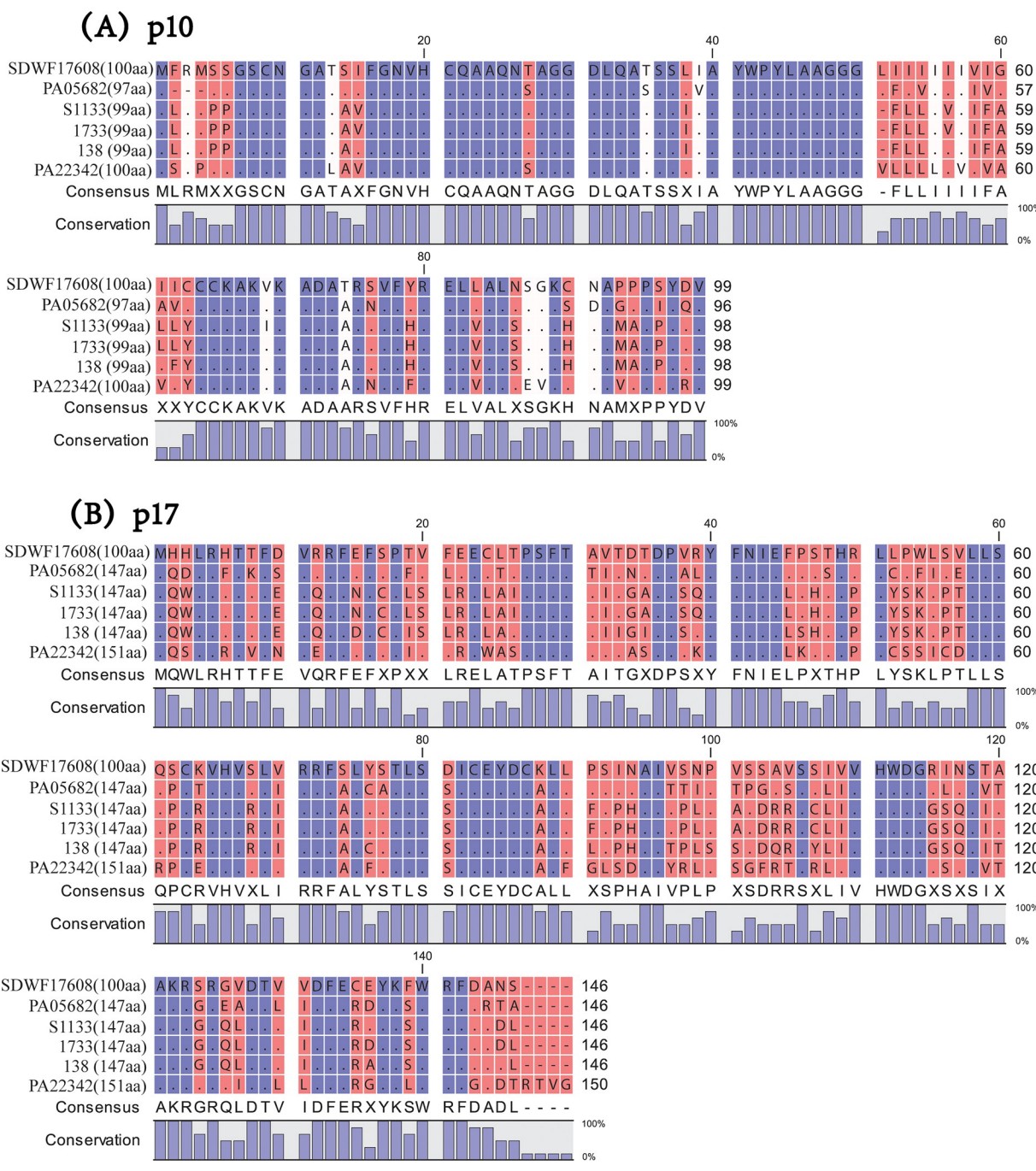

**Fig 2. Amino acid alignment of p10 and p17 protein.** A: The p10 protein of Reo/SDWF /Pheasant/17608/20 align with the homologous protein of Reo/PA/Broiler/05682/12 (or PA05682), S1133, 1733, 138 and Reo/PA/Turkey/22342/13 (or PA22342) strains; B: The p17 protein of Reo/SDWF /Pheasant/17608/20 align with the homologous protein of Reo/PA/Broiler/05682/12 (or PA05682), S1133, 1733, 138 and Reo/PA/Turkey/22342/13 (or PA22342) strains.

29.7–60.2%). In comparisons with different avian species-origin ARVs, the pheasant ARV showed highest identity with PA05682 broiler ARV in μB-, σA-, σB- and σNS-encoding genes (nt: 90.9–99.6%; aa: 97.6–99.7%), high identity with ARV 1733 strain in λA- and μNS-encoding genes (nt: 81.0–91.4%; aa: 92.1–98.5%), and high identity with ARV 138 strain in λB-, λC- and μA-encoding genes (nt: 89.3–91.7%; aa: 96.4–98.1%). The SDWF17608 showed moderate

**Table 2. Sequence identities between the pheasant ARV (Reo/SDWF/Pheasant/17608/20) and reference strains of *Orthoreovirus* genus members.**

| ARV Reference Strain | The pheasant ARV (Reo/SDWF/Pheasant/17608/20) | | | | | | | | | | | | | | | | | | | |
|---|---|---|---|---|---|---|---|---|---|---|---|---|---|---|---|---|---|---|---|---|
| | λA | | λB | | λC | | μA | | μB | | μNS | | σA | | σB | | σC | | σNS | |
| | nt | aa | nt | aa | nt | aa | nt | aa | nt | aa | nt | aa | nt | aa | nt | aa | nt | aa | nt | aa |
| PA05682 | 86.5 | 97.5 | 87.8 | 97.5 | 88.1 | 95.7 | 86.7 | 96.3 | 99.6 | 99.7 | 80.5 | 91.4 | 88.2 | 98.6 | 90.9 | 97.6 | 64.1 | 60.2 | 91.2 | 98.4 |
| S1133 | 91.2 | 98.1 | 83.3 | 96.4 | 72.7 | 84.3 | 88.2 | 97.3 | 84.6 | 94.7 | 80.8 | 91.8 | 90.2 | 96.9 | 85.1 | 94.6 | 54.2 | 49.5 | 81.6 | 93.8 |
| 1733 | 91.4 | 98.5 | 83.5 | 96.7 | 72.9 | 84.2 | 88.3 | 97.4 | 84.6 | 94.7 | 81.0 | 92.1 | 90.3 | 97.8 | 85.3 | 95.1 | 54.3 | 49.2 | 81.8 | 94.0 |
| 138 | 88.9 | 98.3 | 89.3 | 98.1 | 91.7 | 96.4 | 89.9 | 97.0 | 89.5 | 97.3 | 80.3 | 91.8 | 89.9 | 98.1 | 90.0 | 97.6 | 55.7 | 50.8 | 89.1 | 96.7 |
| PA22342 | 84.4 | 96.8 | 83.3 | 95.1 | 83.7 | 90.6 | 86.7 | 96.3 | 84.1 | 93.5 | 80.4 | 90.1 | 88.0 | 97.1 | 71.7 | 77.2 | 55.6 | 51.4 | 78.9 | 91.3 |
| J18 | 77.2 | 94.7 | 76.1 | 91.0 | 70.4 | 79.0 | 74.4 | 87.2 | 75.8 | 89.8 | 71.3 | 81.0 | 76.9 | 92.1 | 64.7 | 68.2 | 42.5 | 29.7 | 78.4 | 90.5 |
| BRoV | 53.8 | 51.0 | 44.5 | 45.2 | 40.1 | 26.0 | 45.8 | 34.3 | 50.8 | 47.0 | 38.6 | 22.0 | 45.7 | 33.3 | 41.7 | 21.7 | NA | NA | 44.6 | 35.5 |
| BRV | 55.0 | 51.5 | 53.6 | 50.2 | 39.7 | 26.5 | 45.8 | 33.6 | 49.6 | 38.8 | 35.8 | 23.4 | 43.0 | 28.9 | 36.0 | 16.9 | NA | NA | 53.6 | 48.9 |
| NBV | 66.1 | 72.7 | 64.6 | 70.5 | 47.2 | 39.3 | 52.6 | 46.7 | 64.0 | 68.0 | 47.8 | 38.8 | 59.6 | 60.8 | 47.9 | 37.6 | 36.7 | 40.2 | 54.1 | 49.5 |
| PuV | 67.0 | 73.5 | 64.2 | 71.0 | 47.8 | 39.3 | 52.7 | 46.8 | 63.5 | 68.7 | 47.9 | 39.0 | 58.5 | 60.2 | 46.5 | 36.5 | 35.8 | 24.5 | 42.5 | 27.1 |
| MRV1 | 49.8 | 43.1 | 55.1 | 54.4 | 41.5 | 27.8 | 44.2 | 28.3 | 51.8 | 45.8 | 41.0 | 23.5 | 43.8 | 29.1 | 40.1 | 19.4 | 34.6 | 19.0 | 44.0 | 23.1 |
| MRV2 | 50.2 | 42.9 | 55.7 | 54.5 | 40.6 | 27.3 | 45.1 | 28.1 | 52.7 | 45.6 | 42.8 | 23.6 | 44.2 | 29.1 | 38.7 | 19.4 | 35.7 | 21.7 | 43.9 | 22.9 |
| MRV3 | 49.9 | 42.7 | 55.4 | 54.4 | 41.2 | 27.3 | 44.6 | 27.9 | 52.3 | 45.6 | 41.7 | 23.5 | 43.5 | 29.1 | 39.6 | 19.2 | 30.6 | 19.0 | 42.2 | 22.6 |

Note: aa = amino acid sequence; nt = nucleotide sequence; NA = sequence not available in Genbank.

identities to PA136491 turkey-derived ARVs (nt: 71.7–88.0%; aa: 77.2–96.8%) and low identity (nt: 64.7–78.4%; aa: 68.2–94.7%) with the J18 duck-origin ARV, and moderate to high identities with other ARV reference strains in the nonstructural protein p10 (nt: 70.1–88.2%; aa: 74.7–96.0%) and p17 (nt: 62.4–86.4%; aa: 61.2–99.3%) which encoded by S1 segment.

The alignment of p10 (Fig 2A) and p17 (Fig 2B) proteins revealed that only p10 protein had a highly conserved region (aa 16 to 50) among the SDWF17608 pheasant ARV and other reference ARV strains.

In comparison with seven non-ARV orthoreovirus strains, the pheasant SDWF17608 shared much lower sequence identities (nt: 30.6–67.0%; aa: 19.0–73.5%) than those within

**Table 3. Comparison of segment 5' and 3' non-coding regions of the pheasant ARV (Reo/SDWF/Pheasant/17608/20) with reference strains of *Orthoreovirus* genus members.**

| Orthoreovirus species | Host | Terminal region sequences | |
|---|---|---|---|
| | | 5' end | 3' end |
| Reo/SDWF/Pheasant/17608/20 | Pheasant | GCUUUU$^{U}$/$_{C}$ | UA$^{U}$/$_{C}$UCAUC |
| Reo/PA/Broiler/05682/12 | Broiler Chicken | GCUUUU$^{U}$/$_{C}$ | UA$^{U}$/$_{C}$UCAUC |
| S1133 | Broiler Chicken | GCUUUUU | UA$^{U}$/$_{C}$UCAUC |
| 1733 | Broiler Chicken | GCUUUU$^{U}$/$_{C}$ | UA$^{U}$/$_{C}$UCAUC |
| 138 | Broiler Chicken | GCUUUUU | UAUUCAUC |
| Reo/PA/Turkey/22342/13 | Turkey | GCUUUUU | UAUUCAUC |
| J18 | Muscovy Duck | GCUUUUU | UA$^{U}$/$_{C}$UCAUC |
| Broome virus (BRoV) | Little Red Flying Fox | GUCAA | UCAUC |
| Baboon orthoreovirus (BRV) | Yellow Baboon | GUAAA | UCAUC |
| Nelson bay virus (NBV) | Grey-headed Flying Fox | GCUUUA | UCAUC |
| Pulau virus (PuV) | Fruit Bat | GCUUUA | UCAUC |
| Mammalian orthoreovirus (MRV)-1 | Mink | GCUA | UCAUC |
| Mammalian orthoreovirus (MRV)-2 | Human | GCUA | UCAUC |
| Mammalian orthoreovirus (MRV)-3 | Masked Civet Cats | GCUA | UCAUC |

ARV species, and their high homology was confirmed to be NBV in λB-, μA-, μB-, σA-, σB-, σC- and σNS-encoding genes (nt: 36.7–64.0%; aa: 37.6–68.0%) and PuV in λA-, λC- and μNS-encoding genes (nt: 47.8–67.0%; aa: 39.0–73.5%).

For UTRs comparison, all ARV strains shared a common motif of first six bases in 5' UTR (Table 3). The seventh base of 5' UTR was conserved in S1133, 138, PA22342 and J18 strains, but showed heterogeneity in other ARV strains, including SDWF17608 strain. The 5' UTR of non-ARV strains was distinct from ARV strains and only showed some conserved regions within species (Table 3). In 3' UTR, the `UAUUCAUC-3'` motif was shared by all ARVs, although the second uracil may be replaced by cytosine base in some segments (Table 3). The last five bases of 3' UTR (`UCAUC-3'`) were shared by all orthoreovirus members (ARVs and non-ARV orthoreoviruses) and considered as the genus common motif.

## Phylogenetic analysis

By the rooted maximum likelihood phylogenetic analysis, the phylogenetic trees of the evolutionary relationships between the pheasant ARV of SDWF17608 and other orthoreovirus members were confirmed. Fig 3 is the phylogenetic trees generated by aligning their nt sequences of the three L (L1-L3) and three M (M1-M3) genome segments and four σ-class genes. For L-class segments analysis, five host-associated groups were formed by the SDWF17608 strain and other reference strains in all three L segments (L1-L3).

All ARV strains were divided into three subclades, chicken ARV, a turkey ARV, and a waterfowl ARV, while all non-ARV strains from mammalian species were more distantly related to ARV strains in a separate branch The SDWF17608 strain was closely related to some historical ARV strains of chicken group I in L1 segment (Fig 3, L1), but clustered with pathogenic ARV strains of chicken group II in L2 and L3 segments (Fig 3, L2 and L3). whereas DAstV-1 formed another clade with pathogenic ARV strains in in L2 and L3 segments.

For M-class segments analysis, host-associated groups were also observed in M1 and M3 trees, but the SDWF17608 strain evolved distant from the grouped ARV strains and fell into

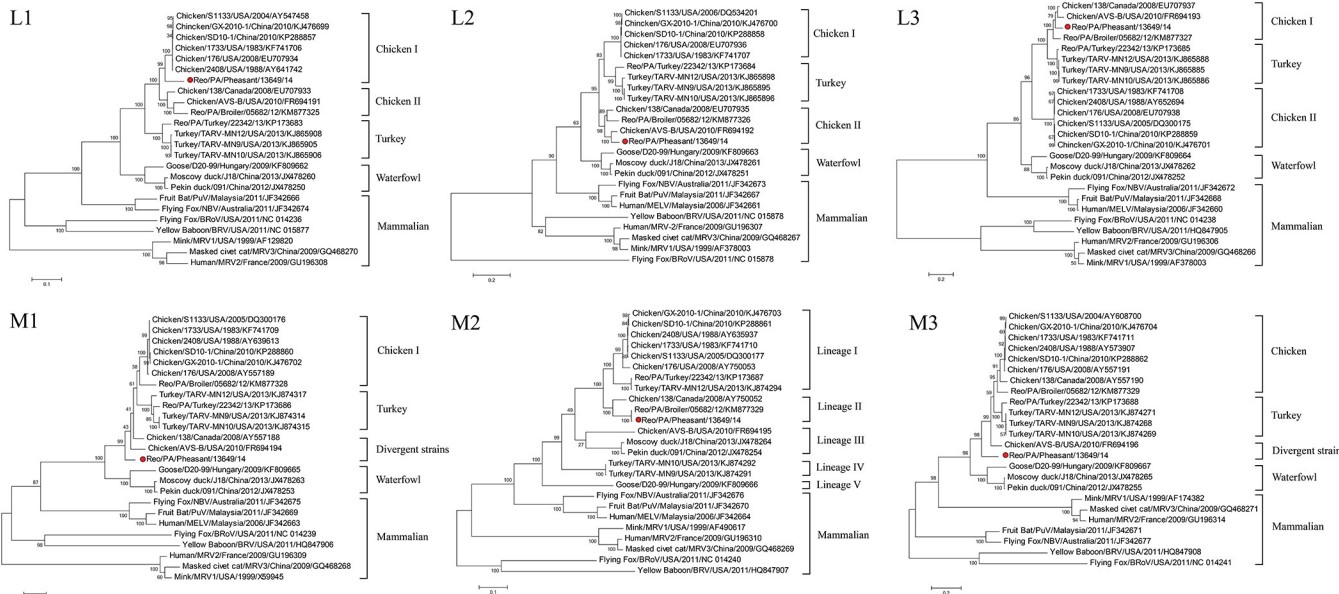

**Fig 3. Phylogenetic trees constructed by avian orthoreovirus (ARV) based on nucleotide sequences of the L-class, M-class and σ-class homologous genome segments or genes.** Note: The Reo/SDWF /Pheasant/17608/20 strain was marked with a red color dot.

the divergent strains group (Fig 3, M1 and M3). The M2 segment of the SDWF17608 strain classified as a lineage 2 member, together with the PA05682 ARV variant and the classic ARV 138 (Fig 3, M2).

The phylogenetic tree analysis based on σ-class gene indicated the variety relationships between SDWF17608 strain and the reference strain. Overall, the ARV and non-ARV orthoreovirus strains were clearly divided into two main branches, and antiretroviral strains can be further divided into distinct subgroups or clusters. (Fig 3).

For σB and σNS genes, SDWF17608 strain evolved distantly from all vaccine strains or classic strains and grouped with the newly emerged pathogenic ARV variants (Fig 3, σB and σNS). In contrast, the σA gene of SDWF17608 strain was not grouped with any classic or pathogenic ARVs, but evolved as a divergent variant strain (Fig 3, σA). As shown in Fig 3, the σC was the most diverse gene among all 10 ARV segment genes.

Phylogenetic analysis of σC genes among all ARV strains resulted in generation of five genotyping clusters, in which the SDWF17608 strain was grouped into genotyping cluster 5 similar to German and Israeli ARV strains [42].

## Whole genome alignment

Whole-genome analysis of pheasant ARV and reference ARV strains was visualized by using the mVISTA online program (Fig 4). Eight of the 10 genome segments of the pheasant ARV and ARV 138 strains were broadly genetically related, with exclusion for the M3 segment and most 3' regions of the S1 segment which corresponding to σC gene. Nevertheless, the M3 and S1 fragments of SDWF17608 strain showed the highest identity to the ARV S1133 and PA05682 strains, respectively. Except for the high similarity in the M2 segment between PA22342 turkey ARV and SDWF17608, most of the other regions are of moderate sequence identity. The duck-origin J18 ARV shared low sequence identities with SDWF17608 strain throughout their genomes, and an even lower identity was observed in S1 segment (<50%), which indicated no segment reassortment between the duck-origin ARV and the pheasants-origin ARV.

## Discussion

ARV as a causative agent of the highly pathogenic and contagious viral arthritis disease of chicken, and ARV-related diseases continue to emerge and expand to other domestic and avian species. The newly emerging ARV field variants mainly affect broiler, layer and turkey productions in the past decade [16, 17, 43, 44].

Numerous studies of the newly emerged ARV field variants suggest that a variety of ARV pathotypes or genotypes were circulating among different flocks and poultry spices [23, 45–47].

Although pheasant production is a relatively small-scale poultry business such as the game birds, pheasants can serve as a potential host repository for ARV transmissions and/or ARV genome reassortments to other avian species. This study provides data for the full-genome characterization of pheasant ARV, a novel pheasant ARV variant. By using the method of pairwise sequence comparison, we confirmed that all 10 genomic segments of SDWF17608 strain were nearly identical to the homologous segments of the reference ARVs.

The conserved regions at the 5' and 3' ends of the positive strand are the same as the ARV members, but there are some differences at the 5' end with the non-ARV reference strains, indicating that the genome segments of SDWF17608 strain were all classified into ARV species and did not undergo fragment reassortment with non-ARV strains [48].

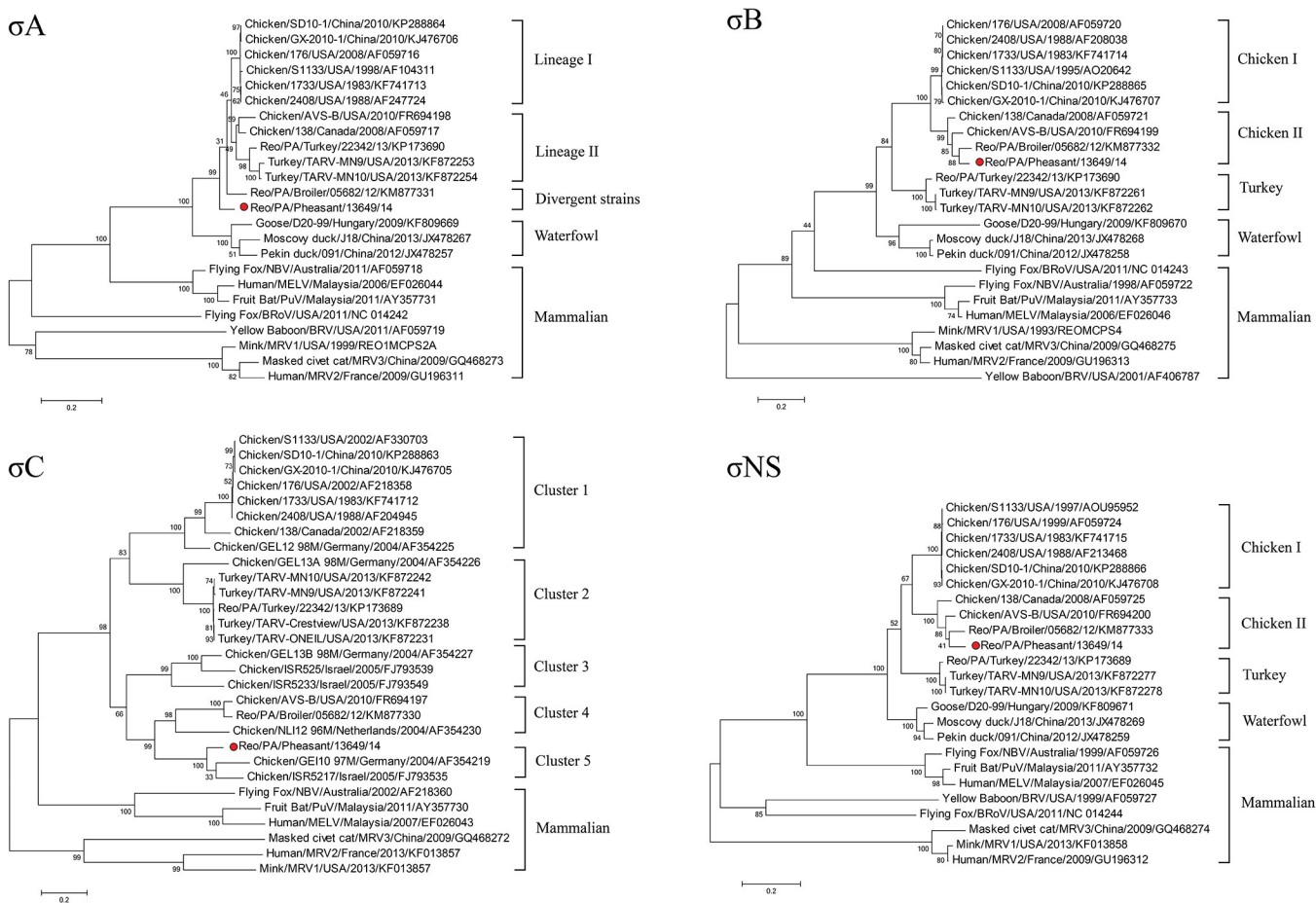

**Fig 4. The mVISTA method for whole genome nucleotide alignment.** Alignment result of the Reo/SDWF /Pheasant/17608/20 in comparisons with the Reo/ PA/Broiler/05682/12 (or PA05682), S1133, 1733, 138, Reo/PA/Turkey/22342/13 (or PA22342) and J18 strains was illustrated; Areas in pink color represent ≥ 90% similarities; and areas in white represent < 90% similarities. The scale bar measures approximate length of the concatenated genome.

The comparison of all coding genes showed that SDWF17608 strain was most closely related to the chicken-origin ARV strains (PA05682 and 138), and the sequence similarity of μB gene with PA05682 broiler antiretroviral strain was the highest (nt: 99.6%; aa: 99.7%). The above results indicated that the PA05682 strain was involved in the ARV genome rearrangement to generate the SDWF17608 strain. As the major outer capsid protein, μB is responsible for virus entry/un-coating and transcriptase activation [49]. The specific sequence of μB gene may be required for an efficient establishment of a successful ARV infection in a specific avian host. The comparison results showed that the SDWF17608 strain was highly identical to the PA05682 broiler antiretroviral μB gene, suggesting that although the SDWF17608 strain was derived from pheasants, it may have the ability to infect other birds. In contrast, σC was the most divergent gene with low sequence identity to the compared strains (nt: <64.1%; aa: <60.2%), which was mainly because the σC gene was the most variable protein in the ARV [50] and showed the less interaction with other viral proteins [42]. In addition, the σB gene encodes the viral in vitro capsid protein and is generally of low homology among different ARV strains. Interestingly, the comparison of σB gene of the SDWF17608 strain in the present study showed host-related identity values, suggesting that the σB gene could be a candidate for a genetic marker to distinguish ARVs from various avian hosts.

The SDWF17608 strain expressed a high degree of diversity with classical ARV strains in its M1, M3 and S2 segments. The SDWF17608 strain was most closely related to PA05682 broiler ARV, and 138 strains were further confirmed by phylogenetic analysis in sequence comparison.

NGS is an important tool for assessing genetic diversity, providing high sequencing coverage for population samples such as viruses [51]. In the present study, the sequencing coverage of SDWF17608 strain was observed from 1390× to 1977× on average for different genomic segments. Such high sequencing coverage allowed us to calculate the reliable numbers of iSNVs in viral genome [52, 53].

In this study, the iSNV information provided us a number of heterogeneous nucleotide positions of the SDWF17608 strain. Six of the 7 iSNV positions were found in the M2 segments of the μB gene, likely due to the location of the μB protein in the outer capsid leading to more immune selection for the μB gene. However, no heterogeneous sites were observed in the σC and σB genes, suggesting a lower level of spontaneous mutation in these two outer capsid proteins in the pheasant ARV strain (SDWF17608).

In conclusion, we have completed the whole-genome sequencing characterizations for this novel pheasant ARV (Reo/SDWF/pheasant/17608/20) through NGS technology on the Illumina Miseq platform. These findings provide scientific genomic data to better understand the genomic evolutionary relationships between different avian species-origin ARVs.

## Supporting information

**S1 Table. Genbank accession numbers of full genome segments of avian orthoreovirus (ARV) reference strains: PA05682, S1133, 138, 1733, 176, J18, BRoV, BRV, NBV, PuV, MRV1, MRV2 and MRV3.**
(DOCX)

## Author Contributions

**Data curation:** Yi Tang, Haiyang Yu, Xiaoning Jiang.

**Formal analysis:** Yi Tang, Xiaoning Jiang.

**Investigation:** Yi Tang, Haiyang Yu, Endong Bao, Dong Wang.

**Methodology:** Yi Tang, Haiyang Yu, Endong Bao, Dong Wang, Huaguang Lu.

**Supervision:** Endong Bao, Dong Wang, Huaguang Lu.

**Visualization:** Endong Bao.

**Writing – original draft:** Yi Tang.

**Writing – review & editing:** Yi Tang, Huaguang Lu.

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
