## [Decision Letter · Decision Letter 0]

15 Feb 2022

PONE-D-21-27563

Genetic characterization of a novel pheasant-origin orthoreovirus using next-generation sequencing

PLOS ONE

Dear Dr. Lu,

Thank you for submitting your manuscript to PLOS ONE. After careful consideration, we feel that it has merit but does not fully meet PLOS ONE’s publication criteria as it currently stands. Therefore, we invite you to submit a revised version of the manuscript that addresses the points raised during the review process.

We look forward to receiving your revised manuscript.

Kind regards,

Islam Hamim, PhD

Academic Editor

PLOS ONE

Journal Requirements:

2. In your Methods section, please provide additional details regarding the animal material used in your study and ensure you have described the source. For more information regarding PLOS' policy on materials sharing and reporting, see " ext-link-type="uri" xlink:type="simple">https://journals.plos.org/plosone/s/materials-and-software-sharing#loc-sharing-materials."

“This research study was supported by Tianjin Ringpu Bio-Technology Co, Ltd., Tianjin, China.”

“The authors have declared that no competing interests exist.”

4. Please include your tables as part of your main manuscript and remove the individual files. Please note that supplementary tables (should remain/ be uploaded) as separate "supporting information" files.

Additional Editor Comments:

Major revision is required. Authors should address to all comments and concerns mentioned by reviewers and revise the paper in response to their advice. If the revision is completed successfully, I will reconsider my decision.

Reviewers' comments:

Reviewer's Responses to Questions

**Comments to the Author**

1. Is the manuscript technically sound, and do the data support the conclusions?

Reviewer #1: Yes

Reviewer #2: Yes

Reviewer #3: Yes

Reviewer #4: No

2. Has the statistical analysis been performed appropriately and rigorously? 

Reviewer #1: No

Reviewer #2: N/A

Reviewer #3: Yes

Reviewer #4: I Don't Know

3. Have the authors made all data underlying the findings in their manuscript fully available?

Reviewer #1: Yes

Reviewer #2: No

Reviewer #3: Yes

Reviewer #4: Yes

4. Is the manuscript presented in an intelligible fashion and written in standard English?

Reviewer #1: Yes

Reviewer #2: No

Reviewer #3: No

Reviewer #4: Yes

5. Review Comments to the Author

Reviewer #1: The Authors investigated an interesting topic concerning the ARV infections. Authors described field strain (Reo/SDWF/Pheasant/17608/20) of avian orthoreovirus (ARV), which have been isolated from a flock of ringneck pheasants in Weifang district of Shandong Province, China. This strain was genetically characterized being a novel field variant strain in conducting whole genome sequencing by using Next-Generation Sequencing (NGS) technique on Illumina MiSeq platform.

The de novo assembly of the ARV reads resulted in generation of 10 ARV-related contigs with the average sequencing coverage from 1390× to 1977× according to 10 ARV genome segments. The complete genomes of this pheasant-origin ARV (Reo/SDWF /Pheasant/17608/20) were 23,495 bp in length and consist of 10 dsRNA segments ranged from 1192 bp (S4) to 3958 bp (L1) encoding 12 viral proteins.

The manuscript was properly structured. The Introduction section was clearly written. The Material and methods section described the ARV isolation from tendon tissue of a pheasant case with hepatopathy symptoms and RT-PCR amplification and Sanger sequencing have been performed. The next step of the studies Next-generation sequencing have been performed. The results section well reported.

However, some changes are necessary before a possible acceptance of the paper.

Here, I am reporting some suggestions aiming to further improve the paper:

Comments:

1. The English grammar needs proofread

2. Line 336, please use the reference

As reoviral infections usually appear in the poultry flocks with an lover immune status, where there is also infection with other viruses that reduce immunity, please mention it and add the references below.

Fowl adenovirus strains 1/A and 11/D isolated from birds with reovirus infection.

Niczyporuk JS, Kozdrun W, Czekaj H, Stys-Fijol N. PLoS One. 2021 Aug 19;16(8):e0256137. doi: 10.1371/journal.pone.0256137. eCollection 2021. PMID: 34411166 Free PMC article.

3. Concerning the Fig. 3 - you have to improve the figure, because is almost impossible to read it.

The data I found interesting and the results connected with full genome sequencing for this novel pheasant ARV were interesting from even the epidemiological point of view and the use of bioinformatic tools to perform NGS analyzes has become useful for determining the evolutionary relationships between the studied ARV strain!

Reviewer #2: The authors report the isolation and genetic characterization of an avian orthoreovirus strain from a pheasant.

Overall, reports about “new” viruses bear always a certain scientific value and are worth to be published. However, the actual manuscript is very poorly written with many grammatical mistakes. The English of the manuscript needs to be improved by someone with high proficiency to target the grammatical errors and to increase the overall language clarity.

Introduction:

The introduction is way too long. Therefore, it shall be shortened and more focused on the scientific objectives of the manuscript.

Variant strain is a redundant term. A virus strain is a variant that possesses unique and stable phenotypic characteristics.

Pheasants, which are the targeted bird species of the manuscript, are not classified as wild birds, but as game birds.

Material Methods:

After building up the readers interest in the introduction highlighting clinical features of the Reovirus infection and the importance of birds other than commercial poultry as reservoirs for ARVs, the manuscript fails to provide information clinical data regarding the infected pheasant: origin, husbandry system, flock size, mortality, morbidity, clinical signs and also post-mortem and histological findings. Why was tendon tissue used for isolation? Did the birds suffered from tenosynovitis?

The term “symptom” is only used in Human Medicine. In Veterinary Medicine the term Clinical Signs is used instead.

Did the clinical signs lasted for 2 weeks?

Was the RNA extracted from the original sample or from ARV propagated in cell culture?

The manuscript should be more clear about the samples used for RNA extraction. The only explanation I can see for the presence of chicken RNA in a pheasant-origin sample is the use of the cell culture propagated ARV for RNA extraction.

Discussion:

The discussion section of this manuscript is very poor. The language is not clear, concise and needs to be corrected grammatically by a proficient speaker.

It is expected that the key-findings of the experiment are summarized and explained in a systematic manner in the context of previous studies leading to deductions and/or hypotheses. The manuscript fails at that, using only selected data for discussion but in confused manner with confusing language.

There is no mention of demarcation criteria for ARV genomes, which is surprising after extensive sequence identity and phylogenetic analysis.

Some references (line 333) might be substituted with a decent review or book chapter.

Lines 352-354: This sentence is not correct. The homology between σB genes is much higher than among σC genes.

References:

The reference list is too long with 57 references. Many of the references should be substituted by a recent review paper or the chapter in Diseases of Poultry, which it was not listed as reference. Additionally, the ICTV chapter on Reoviridae was recently updated and should also be used as reference.

Reviewer #3: PONE-D-21-27563

Title: Genetic characterization of a novel pheasant origin orthoreovirus using next generatio sequencing

Corresponding author: Huaguang Lu

This manuscript summarizes the genetic characterization of a novel Pheasant ARV. While the work is well done and described, there are some concerns about sequencing of segmented viruses and the possibility of mixed isolates. Starting with a pure isolate is a must when you want to characterize a virus (especially when this virus is segmented). In addition, if no info is available on the potential mixed nature of the virus is very difficult to know if the sequences obtained from each of the genes match with the other genes obtained. It wasn’t clear if diverse sequences were obtained for each gene. This situation at least needs to be clarified, explained and discussed in this paper. More information about the origin of the virus is needed and its impacts on the flocks where it was retrieved. The last information will help associate findings with potential use of the information in preventing issues caused by this specific virus. Finally, there are passages of the manuscript where grammar needs to be revised for clarity. Verbal tenses and sentence construction could be improved. Discussion overall could be improved.

ABSTRACT

It is common in reovirus isolates to obtain more than one reovirus or several different viruses including adeno, rota, parvo on the isolates. Was that something that was encountered in your work? If several Reo strains were detected how you know which segments match to form the strain you are molecularly characterizing?

INTRODUCTION

Line 50. Please revise sentence for clarity

Line 54. Check grammar

Some passages require grammar revision

MATERIALS AND METHODS

Lines 99 to 105. Was the virus isolate plaque purified? This would be important if you want to characterize a specific virus. As mentioned above there is a high percent of isolates that possess multiple reoviruses and even other viruses.

Line 129. Should say “was loaded”

NGS doesn’t allow to match the right pieces into an existent Reo gene if there are more than one reovirus present in the sample. That is why is crucial to know if the original isolate was pure (only on reovirus was detected)

Which ARV was used as reference for the construction of the full genome?

Line 161 to 162. Not sure about this, what is the meaning that this genome was set as a scale?

What is the reference helping with?

Line 167 to 171. Was this sequencing done in China or the US and, are there any animal experiments involved? Why there is an IACUC approval?

RESULTS

Lines 175 to 181. Please review sentences and check grammar. In its present form is difficult to understand. In addition, be clear with the results that are being shown. In addition to GenBank numbers refer to relevant data to compare homologies

Images are extremely low quality, they are very difficult to read, please correct

Were all sequences retrieved from each gene similar? If so, how similar they were in terms of homologies?

Figure 4 is not in the manuscript. I will need in order to review this section of the manuscript

DISCUSSION

Line 336 should say “species” says “spices”

Discussion is somewhat unorganized and do not cover the issue of full genome sequencing segmented viruses.

Reviewer #4: Tang et al. grew a virus from an infected pheasant in a cell line of avian origin, sequence the resulting S1 gene by Sanger sequencing and then the whole genome by Illumina. The obtained sequence is compared to available sequences to conclude that this virus has some gene segment rearrangements.

General comment: The study is of little interest as the sequence is from just one animal and the virus was not directly sequenced, but first grown in a cell line, so the virus sequenced might differ from the original one. The results obtained are of little, if any, interest. Also, it is beyond my understanding why they grow the virus, make DNA copies of the genome segment S1 and perform Sanger sequencing and afterwards go to Illumina for the rest 9 segments while Sanger is more accurate: what is the reason?

Particular observations:

- In the “Introduction” section, the taxonomical data is out of date: the references given are 20 years old at least and in fact there are more than 5 species in the Orthoreovirus genus.

- If they want to be accurate in protein listing, there are some missing proteins that are formed inside the infected cell by post-translational processing (muBC, muBN...).

- No description at all is given of the method used for virus isolation. Also, the antibody used is not described, it is instead said that it was characterized on a previous study, but the same reference is given on such study: it should at least be said if the antibody was raised against the whole, virus, a particular protein, group of proteins, etc.

- Fig 1 is prescindible: just data from their methodology that is not either informative or useful at all.

- Fig 2 Why this comparison of just p10 and p17? Why not then sigmaC that is supposed to be the most variable? I don´t see any reason for this figure neither.

- The references are just a nonsense: there are many missing articles. Just to cite some, those describing the tricistronic nature of the S1 gene (discussion of that deserves figure 2 to the authors). On the other hand, several others are no cited on the text, 35 for example, something that makes sense as it has nothing to do with the subject of this paper.

6. PLOS authors have the option to publish the peer review history of their article (what does this mean?). If published, this will include your full peer review and any attached files.

Reviewer #1: No

Reviewer #2: No

Reviewer #3: No

Reviewer #4: No

---

## [Author Response · Author response to Decision Letter 0]

18 Apr 2022

We have completed our revision accordingly for resubmission, all of the revised parts throughout the manuscript were highlighted yellow. Thank you for considering this manuscript for publication by PLOS ONE.

---

## [Decision Letter · Decision Letter 1]

1 Aug 2022

PONE-D-21-27563R1

Genetic characterization of a novel pheasant-origin orthoreovirus by next-generation sequencing

PLOS ONE

Dear Dr. Huaguang Lu,

Thank you for submitting your manuscript to PLOS ONE. After careful consideration, we feel that it has merit but does not fully meet PLOS ONE’s publication criteria as it currently stands. Therefore, we invite you to submit a revised version of the manuscript that addresses the points raised during the review process.

If applicable, we recommend that you deposit your laboratory protocols in protocols.io to enhance the reproducibility of your results. Protocols.io assigns your protocol its own identifier (DOI) so that it can be cited independently in the future. For instructions see: https://journals.plos.org/plosone/s/submission-guidelines#loc-laboratory-protocols. Additionally, PLOS ONE offers an option for publishing peer-reviewed Lab Protocol articles, which describe protocols hosted on protocols.io. Read more information on sharing protocols at https://plos.org/protocols?utm_medium=editorial-emailutm_source=authorlettersutm_campaign=protocols.

We look forward to receiving your revised manuscript.

Kind regards,

Islam Hamim, PhD

Academic Editor

PLOS ONE

Journal Requirements:

Additional Editor Comments (if provided):

Authors need to address following concern from a reviewer-----

"The author used a virus isolated from tendons to perform sequencing and genetically characterize this virus. If plaque purification was not done a chance of mixing sequences from different viruses is a possibility. I still don't see comments and discussion related to this. Please highlight and properly point lines where this topic is addressed"

Reviewers' comments:

Reviewer's Responses to Questions

**Comments to the Author**

1. If the authors have adequately addressed your comments raised in a previous round of review and you feel that this manuscript is now acceptable for publication, you may indicate that here to bypass the “Comments to the Author” section, enter your conflict of interest statement in the “Confidential to Editor” section, and submit your "Accept" recommendation.

Reviewer #1: All comments have been addressed

Reviewer #3: (No Response)

Reviewer #5: All comments have been addressed

2. Is the manuscript technically sound, and do the data support the conclusions?

Reviewer #1: Yes

Reviewer #3: Yes

Reviewer #5: Yes

3. Has the statistical analysis been performed appropriately and rigorously? 

Reviewer #1: N/A

Reviewer #3: Yes

Reviewer #5: Yes

4. Have the authors made all data underlying the findings in their manuscript fully available?

Reviewer #1: Yes

Reviewer #3: Yes

Reviewer #5: Yes

5. Is the manuscript presented in an intelligible fashion and written in standard English?

Reviewer #1: Yes

Reviewer #3: Yes

Reviewer #5: Yes

6. Review Comments to the Author

Reviewer #1: Authors described field avian orthoreovirus strain (Reo/SDWF/Pheasant/17608/20) isolated from a flock of game pheasants in Weifang, Shandong Province. The next-generation sequencing (NGS) technique on the Illumina MiSeq platform to determine the sequence of reovirus strain have been used. Among 870,197 35–151-mer sequencing reads, 297,711 (34.21%) were ARV sequences. The complete genome of SDWF17608 of pheasant origin was 23,495 bp in length and comprised 10 dsRNA segments ranging from 1192 bp (S4) to 3958 bp (L1) encoding 12 viral proteins. The sequence comparison between the current strain and classic ARV reference strains revealed 58.1–100% nucleotide (nt) identities and 51.4–100% amino acid (aa) identities in genome coding genes. The 10 RNA segments had conserved termini at the 5’ (5’- 33 GCUUUU) and 3’ (UCAUC-3’) ends, which were identical to those of published ARV strains. Phylogenetic analysis revealed that the SDWF17608 pheasant ARV field strain was closely related to chicken ARV strains in seven genome segment genes but showed significant sequence divergence in the M1, M3, and S2 segments. These findings suggested that this field variant of pheasant origin was a divergent ARV strain and likely originated from the reassortment of different chicken ARV strains.

The above work presents a detailed molecular (nucleotide and amino acid) characterization of reovirus strains isolated from pheasants using NGS sequencing. The prepared figures and tables presented in the above work are clear, understandable and concise and essential and significantly enrich the content of the discussed work.

These studies and the results obtained will allow for better understanding / illustrating the evolution of the genome of ARV strains and phylogenetic connections that may occur in strains of reoviruses isolated from different species of birds. The work has a high cognitive value and applies for its publication in the PLOS one journal.

Reviewer #3: The author used a virus isolated from tendons to perform sequencing and genetically characterize this virus. If plaque purification was not done a chance of mixing sequences from different viruses is a possibility. I still don't see comments and discussion related to this. Please highlight and properly point lines where this topic is addressed.

Reviewer #5: (No Response)

7. PLOS authors have the option to publish the peer review history of their article (what does this mean?). If published, this will include your full peer review and any attached files.

Reviewer #1: **Yes: **Jowita Samanta Niczyporuk

Reviewer #3: No

Reviewer #5: No

---

## [Author Response · Author response to Decision Letter 1]

16 Sep 2022

Thank you for providing us with detail instructions and the reviewers’ comments to revise the manuscript. We have completed the reversion (R2) accordingly for re-submission. Thank you for considering this manuscript for publication by PLOS ONE.

---

## [Decision Letter · Decision Letter 2]

27 Oct 2022

Genetic characterization of a novel pheasant-origin orthoreovirus by next-generation sequencing

PONE-D-21-27563R2

Dear Dr. Huaguang Lu,

We’re pleased to inform you that your manuscript has been judged scientifically suitable for publication and will be formally accepted for publication once it meets all outstanding technical requirements.

Kind regards,

Islam Hamim, PhD

Academic Editor

PLOS ONE

Additional Editor Comments (optional):

Accepted

Reviewers' comments:

Reviewer's Responses to Questions

**Comments to the Author**

1. If the authors have adequately addressed your comments raised in a previous round of review and you feel that this manuscript is now acceptable for publication, you may indicate that here to bypass the “Comments to the Author” section, enter your conflict of interest statement in the “Confidential to Editor” section, and submit your "Accept" recommendation.

Reviewer #5: (No Response)

2. Is the manuscript technically sound, and do the data support the conclusions?

Reviewer #5: (No Response)

3. Has the statistical analysis been performed appropriately and rigorously? 

Reviewer #5: (No Response)

4. Have the authors made all data underlying the findings in their manuscript fully available?

Reviewer #5: (No Response)

5. Is the manuscript presented in an intelligible fashion and written in standard English?

Reviewer #5: (No Response)

6. Review Comments to the Author

Reviewer #5: (No Response)

7. PLOS authors have the option to publish the peer review history of their article (what does this mean?). If published, this will include your full peer review and any attached files.

Reviewer #5: No
